# Recent Research Agendas in Mining Equipment Management: A Review

**Shi Qiang Liu** [1,*] **, Zhaoyun Lin** [1]**, Debiao Li** [1]**, Xiangong Li** [2]**, Erhan Kozan** [3] **and Mahmoud Masoud** [4]

1   School of Economics and Management, Fuzhou University, Fuzhou 350116, China
2   School of Mining Engineering, China University of Mining and Technology, Xuzhou 221116, China
3   School of Mathematical Sciences, Queensland University of Technology, Brisbane 4000, Australia
4   Centre for Accident Research and Road Safety, Queensland University of Technology, Brisbane 4000, Australia
*   Correspondence: samsqliu@fzu.edu.cn

**Abstract:** Nowadays, with the advancement of technological innovations and wide implementation of modern mining equipment, research topics on mining equipment management are attracting more and more attention from both academic scholars and industrial practitioners. With this background, this paper comprehensively reviews recent publications in the field of mining equipment management. By analysing the characteristics of open-pit mine production and haulage equipment types, problem definitions, formulation models and solution approaches in the relevant literature, the reviewed papers are classified into three main categories, i.e., shovel–truck (ST); in-pit crushing–conveying (IPCC); and hybrid IPCC-ST systems. Research progress and characteristics in each categorized mining equipment system are discussed and evaluated, respectively. With a thorough assessment of recent research agendas, the significance of developing state-of-the-art mining equipment scheduling/timetabling methodologies is indicated, based on the application of classical continuous-time machine scheduling theory. Promising future research directions and hotspots are also provided for researchers and practitioners in the mining industry.

**Keywords:** open-pit mining; mining equipment management; operations research in natural resources; shovel–truck; in-pit crushing–conveying; research opportunities

## 1. Introduction

Nowadays, with the rapid development of modern mining technology, semi-automated or automated machinery and equipment have been widely applied in a variety of mine sites around the world. A contemporary mine site typically lasts from many years to several decades, continually providing metallic ores that are important raw materials for the manufacturing industry or non-metallic ores that are also vital to other industries such as construction, agriculture and chemical industries. For mineral-rich countries (e.g., Australia, Canada, Russia, Chile, Iran), the mining sector creates millions of jobs and substantial export earnings which are sources of national wealth to drive the development of other economic sectors such as education, transportation and commerce. On the other hand, mining exploration and exploitation require a large capital investment and involve huge annual cash flows. Therefore, many researchers have studied different kinds of mining optimisation problems from different perspectives to maximize the value of the whole mining process under constraints such as resource capacity, precedence, extraction, haulage, crushing, grade control, stockpiling, railing, shipment, environmental protection and economic issues. Among these studies in mining optimisation, some were devoted to modelling the ultimate mine design and long-term strategic planning problems over the life of a mine (with the time horizon of 10–30 years, typically); the majority of works focused on open-pit mine block sequencing problems at the tactical level (with the time horizons measured in months); some focus on short-term mine equipment planning and scheduling problems (with time windows measured in weeks) at the operational level.

As far as it is known, there have been a range of literature review papers in the field of mining management, as listed in the following:

- a review of solution methodologies for the problem of open-pit mine production scheduling by Fathollahzadeh et al. [1];
- a comprehensive interdisciplinary review of mine supply chain management by Zeng et al. [2];
- a systematic review of machine learning applications in mining exploration, exploitation and reclamation by Jung and Choi [3];
- a literature review of in-pit crushing–conveying (IPCC) technology in open-pit mining operations by Osanloo and Paricheh [4];
- a survey of modelling the integrated mine-to-client supply chain by Leite et al. [5];
- a review of deep learning in mining and mineral processing operations by Fu and Aldrich [6];
- a review of game theory for analysing and improving environmental management in the mining industry by Collins and Kumral [7];
- a review of short-term planning for open-pit mines by Blom et al. [8];
- a review of models and algorithms on fleet management systems for mining by Moradi Afrapoli and Askari-Nasab [9];
- a review of equipment selection for surface mining by Burt and Caccetta [10];
- a review of soft computing technology applications in some mining problems by Jang and Topal [11];
- a review of real-time optimisation in underground mining production by Song et al. [12];
- a review of optimized open-pit mine design and pushbacks by Meagher et al. [13];
- a library of open-pit mining problems and benchmark instances (MineLib) by Espinoza et al. [14];
- a classification and literature review of operations research for mining by Kozan and Liu [15];
- a review of operations research in mine planning by Newman et al. [16];
- a review of models and algorithms for long-term open-pit mine production planning by Osanloo et al. [17];
- a review of critical parameters for sizing equipment in open-pit mining by Bozorgebrahimi et al. [18];
- an overview of solution strategies used in truck dispatching systems for open-pit mines by Alarie and Gamache [19];

Despite the existence of these literature review papers for the mining industry, it is still rare to find a comprehensive and up-to-date literature review focusing on mining equipment (e.g., shovels/excavators, trucks, crushers, and conveyors) management at the operational level. To fill this research gap, this paper aims to summarize the recent research progress in the field of mining equipment management. By assessing over 100 recent papers published in leading or non-mainstream journals in different disciplines including operations research, management science, computer science, mathematics, artificial intelligence, transportation research, resources policy, mining engineering, science and technology, we classify the current research agendas on mining equipment management into three main categories, namely, shovel–truck (ST); in-pit crushing–conveying (IPCC); and hybrid IPCC-ST systems. Based on such a classification, we also discuss emerging and promising research opportunities in the arena of mining optimisation.

The remainder of the paper is outlined as follows. Section 2 will present an overview of recent publications relevant to the shovel–truck (ST) system. Section 3 will review the recent papers on the in-pit crushing–conveying (IPCC) system. The studies of the hybrid IPCC-ST system will be summarized in Section 4. Potential opportunities are discussed in Section 5. The last section concludes this paper.

## 2. Shovel–Truck (ST) System

In open-pit mining, shovels (excavators) and trucks are the most widely used equipment, because material handling (mainly excavation with haulage) is the most important mining operation. According to previous studies, material handling accounts for nearly 50% of the total operating cost in most open-pit mines. In addition, excavation and haulage operations are highly interdependent and inter-reliant. Usually, a fleet of mining trucks is compatibly matched with a large shovel; and the productivity (e.g., reducing the total idle time) of one shovel must rely on the truck fleet management (e.g., optimising the cyclic queuing times of a truck fleet). For better understanding, the main components and operation processes of the ST system are illustrated in Figure 1.

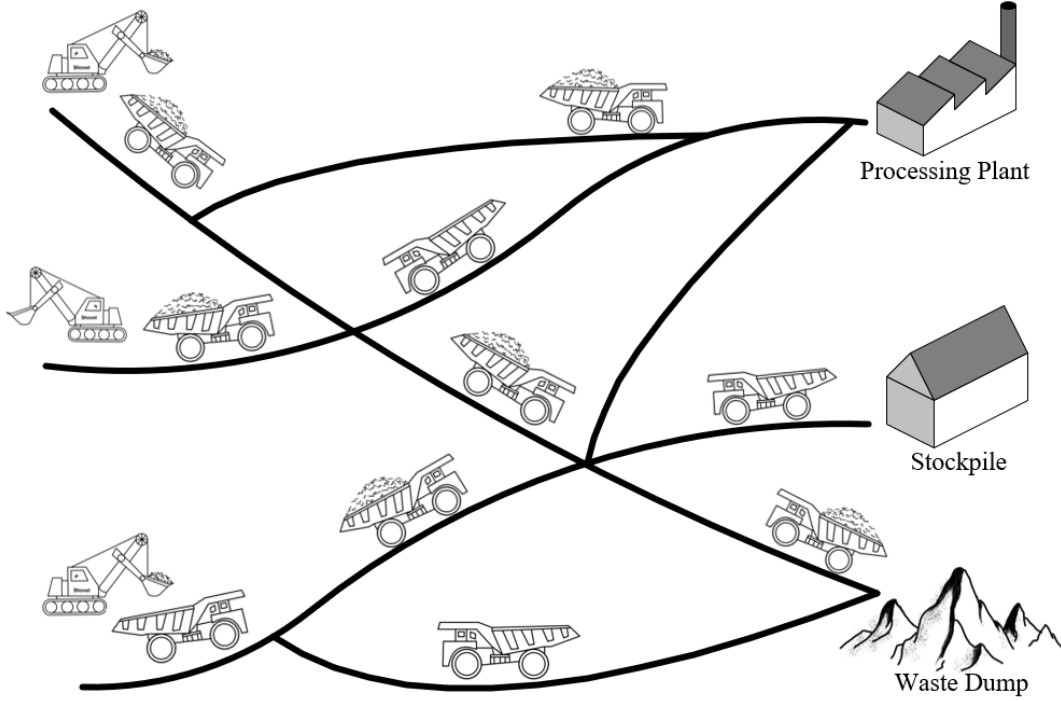

**Figure 1.** Illustration of main components and operation process of the ST system.

Due to huge investment in modern mining equipment, mining companies are keen to improve efficiency and productivity of their ST systems. In the following, an overview of recent research papers relevant to management of open-pit ST system is presented in chronological order.

Young and Rogers [20] devised a high-fidelity modelling (HFM) method to analyse cascading behaviours of the run-of-mine material haulage during and after dumping. The proposed HFM is useful to calibrate the simulation process of dumping trucks to better match reality in open-pit mines.

Liu et al. [21] introduced a new short-term operational-level mine excavators timetabling (MET) problem, which aims to determine the optimal timetable (i.e., starting and completion times) of excavators with the application of continuous-time machine scheduling and disjunctive graph theory. The objective of the MET is to minimize the total weighted tardiness (delay cost) and the total weighted sequence-dependent movement time (relocation cost). The mixed integer programming (MIP) model with the use of IBM ILOG-CPLEX, a construction algorithm and a hybrid tabu search–threshold accepting metaheuristic algorithm were developed to solve small-, middle- and large-size MET instances.

De Carvalho and Dimitrakopoulos [22] developed a deep Q-learning reinforcement learning (RL) method for the truck dispatching problem with adherence to the operational plan and fleet utilisation in a copper–gold mining complex. By training and simulating

uncertainties in geological attributes and equipment performance, the proposed DL method can bring about improvements on truck fleet management.

Upadhyay et al. [23] presented a short-term production scheduling model which allocates shovels over continuous time frames while satisfying capacity utilisation of equipment, meeting blending targets and adhering to the strategic-level plans. A case study of the model was conducted for an iron ore open-pit mine.

Aguayo et al. [24] analysed the potential productivity and safety benefits with the incorporation of a surge loader into the loading and haulage systems in open-pit mining. The interaction analysis indicated that the incorporation of a surge loader in the ST system leads to the reduction likelihood of truck-overfilling and uneven loading.

Elijah et al. [25] developed a queuing theory model to calculate the inter-arrival and service rates with different numbers of trucks and shovels. Their proposed model was coded in MATLAB software and applied to a limestone surface mine. The result analysis showed that after reaching an optimal matching point, increasing the number of trucks may reduce the productivity of the ST system and lead to a rise in the cost per ton of material.

Wang et al. [26] implemented a regression analysis method to assess the fuel consumption patterns of different types of mining trucks with the consideration of multi-dimensional characteristics such as load transport distance, lifting height, and operation time per cycle. Their experimental results indicated that the load, lifting height, haulage distance and queuing status are the key indicators influencing fuel consumption of mining trucks.

Bakhtavar and Mahmoudi [27] proposed a scenario-based robust optimisation (SBRO) method to solve the shovel–truck allocation (STA) problem in open-pit mines. A two-stage STA mathematical programming model with uncertainties was developed. In the first stage of SBRO, uncertainty in shovel and crusher capacity is considered to minimize the trucks' cycle and transportation cost. The second stage aims to minimize the total number of trucks by considering the availability of trucks for each shift. Compared to the traditional scheme, the proposed SBRO can increase the throughput of the ST system and thus reduce the operating cost measured in per ton of ore.

Basiri et al. [28] performed a reliability analysis of mining shovels to measure the risk of failures. They applied statistical methods to generate fault fitness functions, sort the importance of subsystems, and determine the shovels' key indicators. Through the application of reliability and risk analysis to assist in making decisions, the productivity of the ST system is improved.

Zhang et al. [29] proposed a multi-objective unmanned truck scheduling model by minimizing transportation cost, waiting time and grade deviation. They developed a hybrid metaheuristic (i.e., decomposition-based constrained dominance principle non-dominated sorting genetic algorithm) to obtain the Pareto optimal schedule of unmanned mining trucks. The obtained solution can considerably reduce the mining cost by decreasing the trucks' waiting time and ore content fluctuation degree.

Kansake and Frimpong [30] presented an analytical model to estimate tire dynamic forces on haul roads. The estimates of tire impact forces can be used as the input for the design of haulage roads.

Shah and Rehman [31] introduced a MIP model to formulate a shovel–truck allocation problem for a cement quarry mine. The results showed that the proposed MIP model can lead to a significant cost reduction and a better coordination in the ST system.

Ozdemir and Kumral [32] developed a two-stage dispatching system to optimise the shovel–truck operations in a multi-pit surface mine. In the first stage, the truck fleets are divided into sub-fleets to work on each specific pit by a simulation-based method. In the second stage, the trucks are simultaneously allocated to each shovel in each pit by the application of linear programming optimisation. The proposed simulation-based optimisation approach has great potential to improve mining productivity.

Dabbagh and Bagherpour [33] employed an ant colony optimisation (ACO) metaheuristic algorithm and a simulation model to determine and analyse the matching factor and applicability comparison, respectively, according to the distribution functions which

relate to the time cycles of mining trucks in heterogeneous transportation fleets in an open-pit mining site. Through a simulation analysis, the production capacities of iron ore and waste dumps can be increased by 4.4% and 4.1%, respectively.

Liu and Chai [34] formulated a MIP model to optimise the route of trucks in an open-pit mine with real-world constraints such as driving distance, traffic density, road capacity, pattern recognition and surface estimation. An improved genetic algorithm (GA) was developed to solve the truck routing problem for minimizing the time-varying energy consumption under the influence of resistance fluctuation.

Moniri-Morad et al. [35] developed a capacity analysis framework to analyse the impact of mining equipment performance on the nominal capacity of haulage fleet. They used the discrete event simulation (DES) model with probabilistic risk assessment to evaluate the effect of different risk levels for enhancing the equipment availability and mitigating the maintenance cost.

Sun et al. [36] applied machine learning techniques such as *k*-nearest neighbour (KNN), support vector machine (SVM) and random forests (RF) to predict the real-time link travel time of open-pit trucks on fixed and temporary roads. Taking a road section as the minimum prediction unit, prediction accuracy was evaluated by the average absolute deviation. The results showed that the proposed prediction model in this study is better than traditional prediction methods in the literature.

Patterson, Kozan and Hyland [37] proposed a novel MIP model for scheduling haulage activity to minimize the shovel–truck energy consumption and meet production targets. Due to the NP-harness, a constructive algorithm and a tabu search metaheuristic were developed to efficiently solve the problem for practical use. To validate the proposed formulation model and solution techniques, an operating mine in southeast Queensland was used as a case study with sensitivity and scenario analysis where significant potential for improvement was found.

Bajany et al. [38] presented a MIP model for the shovel–truck dispatching problem with the objective of minimizing the fuel consumption while satisfying the handling demand of dump sites. The results showed that the average litre of fuel consumption per ton of mineral haulage could be significantly reduced by 4.64% by the proposed ST dispatching optimisation model.

Baek and Choi [39] studied open-pit transportation road design to support efficient truck haulage operations. In this study, the optimal boundary of an open-pit workbench was designed; the raster-based least-cost path analysis was used to generate a two-dimensional road layout; the road layout was altered to improve the stability by considering the radius of curvature; and the proposed method facilitated the existing mine design software to improve the haulage road design in open-pit mines.

Dindarloo and Siami-Irdemoosa [40] conducted a pioneering study of pattern recognition and failure forecasting on mining equipment with the application of data mining methods. Based on historical failure/overhaul data on shovels, the *K*-means clustering algorithm was used to identify the fault type and support vector machine (SVM) was used to predict the imminent failure.

Burt et al. [41] developed a MIP model to formulate a multi-location multi-period equipment selection problem, which involves choosing an appropriate fleet of trucks and loaders in a multi-location and multi-dumpsite open-pit mine. The underlying equipment selection problem was transformed into a kind of multi-commodity multi-period network flow optimisation problem and thus the corresponding solution approaches can be adapted to solve large-scale instances. They provided the calculation formulas that can accurately determine the cost variations when equipment moves from one phase to the next in a period.

Chang et al. [42] studied an open-pit truck scheduling problem with the consideration of different transportation costs and revenues. A MIP model is developed to formulate the problem and analyse the problem properties such as loading points. Based on the analysis of properties and upper bounds, a heuristic algorithm with two improvement strategies is developed to solve the open-pit truck scheduling problem.

Dindarloo et al. [43] developed a stochastic simulation framework for truck and shovel selection and sizing in open-pit mining. Uncertainties of the underlying material loading and haulage parameters were defined and built into the stochastic model. A discrete event simulation was employed to simulate the stochastic material handling process in a case study.

Rodrigo et al. [44] proposed an availability-based simulation-and-optimisation framework for truck allocation in open-pit mines. Considering the reliability, availability and maintainability of mining equipment, the truck fleet was allocated according to the route to improve the availability and productivity of the ST system.

Choi and Nieto [45] investigated the haulage routing of off-road trucks in construction and mining sites by the application of a modified least-cost path algorithm with an embedded 3D render window of Google Earth. Thus, a so-called Google-Earth-based optimal haulage routing system (GEOHARTS) was developed to provide a rational solution to support the truck haulage operations in an open-pit coal mine.

Souza et al. [46] developed a hybrid algorithm that combines merits of two metaheuristics, i.e., general variable neighbourhood search (GVNS) and greedy randomized adaptive search procedures (GRASP), to find the optimal dynamic truck allocation solution that minimized the cost deviation (with a gap of less than 1%) and the number of trucks in a short time, with the consideration the compatibility and capacity of trucks and shovels.

Topal and Ramazan [47] developed a MIP model to formulate a mine equipment scheduling model with the objective of minimizing the total maintenance cost. The proposed model aimed to achieve annual production targets by producing an optimum schedule of truck fleets over a multi-year time horizon.

Choi et al. [48] combined multi-criteria evaluation and least-cost path analysis to develop a software called Dump Traveller. The software considered factors related to the efficiency and safety of the mining haulage process, determined the weights of indicators by pairwise comparison, generated the optimal route for mining truck fleets, estimated the travel times in truck dispatching, and helped decision makers make a better balance between road maintenance and traffic jams.

Ercelebi and Bascetin [49] proposed a mining trucks allocation model by the theory of closed queueing network and a shovel–truck dispatching model by linear programming. The proposed models provided the capability of estimating performance measures (e.g., extracting throughput, optimum number of trucks, mean waiting time of trucks, optimum dispatching policy, and the cost of ore haulage) of an open-pit ST system.

Table 1 summarises the main characteristics of recent papers on the ST system in terms of the authors, publication year, journals, country of the first author, problem types and solution techniques. As shown in Table 1, some findings are given as follows. First, most research considered the mixture of shovels and trucks, e.g., determining the best matching factor; selection with sizing of trucks and shovels; dispatching a fleet of trucks to one shovel. In comparison, investigation of individual shovel or truck management system is rare relatively. Second, most of studies on the ST system belong to a kind of the planning-type optimisation problems such as the ST allocation/dispatching/assignment/matching problem. In contrast, few studies focused on more complicated scheduling-type problem based on the application of classical machine scheduling theory. Note that planning deals with the optimisation problems of resource capacity, facility design, equipment allocation and personnel deployment without considering timing factors. Scheduling is concerned with the efficient allocation of equipment units to jobs (operations) and sequencing the operations on each equipment unit with timing factors. For example, the parallel-machine scheduling with sequence-dependent set-up times was recently applied to a real-world mine excavators timetabling case [21]. Indeed, the dynamic vehicle routing problem could be applied to the routing optimisation of open-pit truck fleets [50,51]. Third, most solution techniques for the ST problems are mainly based on the formulation of MIP models with the use of exact MIP solvers. More efficient solution approaches, such as metaheuristic algorithms, which can efficiently solve large-scale instances, are relatively occasional.

Finally, for scheduling (dispatching and sequencing) a fleet of trucks associated with a shovel, most existing mathematical programming models are relatively basic. To be more applicable in practice, the ST scheduling models should be extended by considering more actual requirements, such as the best matching factor, the selection of trucks/shovels, the layout of haulage roads, the queuing (e.g., waiting/idle times) of trucks in the scheduling process, and maintenance/failure of mining equipment, etc.

**Table 1.** Characteristics analysis of publications on the shovel–truck (ST) system [20–49].

| Authors | Year | Country | Problem Types | Solution Techniques |
|---|---|---|---|---|
| Young and Rogers | 2022 | USA | Mine haul truck dumping process simulation | A high-fidelity modelling method |
| Liu et al. | 2022 | China | Mine excavators timetabling | Mixed integer programming and metaheuristics |
| de Carvalho and Dimitrakopoulos | 2021 | Canada | Integrated truck-dispatching and production | Reinforcement learning |
| Upadhyay et al. | 2021 | Canada | Production scheduling with shovel allocation | Mixed integer programming |
| Aguayo et al. | 2021 | Chile | Productivity and safety of shovel–truck system | Interaction analysis |
| Elijah et al. | 2021 | Kenya | Shovel–truck haulage optimisation | Queuing theory |
| Wang et al. | 2021 | China | Mine truck fuel consumption analysis | Regression analysis |
| Bakhtavar and Mahmoudi | 2020 | Iran | Shovel–truck allocation | Scenario-based robust optimisation |
| Basiri et al. | 2020 | Iran | Reliability assessment of shovel–truck system | Statistical methods |
| Zhang et al. | 2020 | China | Multi-objective unmanned truck scheduling | Improved genetic algorithms (NSGA-II) |
| Kansake and Frimpong | 2020 | USA | Estimate tire dynamic forces on haul roads | An analytical model |
| Shah and Rehman | 2020 | Pakistan | Shovel–truck allocation problem | Mixed integer programming |
| Ozdemir and Kumral | 2019 | Canada | A two-stage shove-truck dispatching system | A simulation-based optimisation approach |
| Dabbagh and Bagherpour | 2019 | Iran | Matching factor of shovel–truck system | Ant colony optimisation |
| Liu and Chai | 2019 | China | Routing optimisation of open-pit trucks | Mixed integer programming |
| Moniri-Morad et al. | 2019 | Iran | Capacity analysis of shovel–truck system | Discrete event simulation |
| Sun et al. | 2018 | China | Prediction of travel times of trucks | Machine learning techniques |
| Baek and Choi | 2017 | Korea | Design of a haul road for an open-pit mine | Douglas–Peucker algorithm |
| Dindarloo and Siami-Irdemoosa | 2017 | USA | Classification and clustering of shovels failures | Data mining techniques |
| Patterson, Kozan and Hyland | 2017 | Australia | Energy efficient shovel–truck scheduling | Mixed integer programming and metaheuristics |
| Bajany et al. | 2017 | South Africa | Shove-truck dispatching | Mixed integer programming |
| Burt et al. | 2016 | Australia | Mining equipment selection | Mixed integer programming |
| Chang et al. | 2015 | China | Open-pit truck scheduling | Mixed integer programming |
| Dindarloo et al. | 2015 | USA | Truck and shovel selection and sizing | Stochastic simulation |
| Rodrigo et al. | 2013 | France | Dynamic open-pit mine truck allocation | Simulation-and-optimisation framework |
| Choi and Nieto | 2011 | Korea | Haulage routing optimisation of mining trucks | Least-cost path algorithm with Google Earth |
| Souza et al. | 2010 | Brazil | Dynamic truck allocation in open-pit mining | Hybrid metaheuristic algorithms |

**Table 1.** *Cont.*

| Authors | Year | Country | Problem Types | Solution Techniques |
|---|---|---|---|---|
| Topal and Ramazan | 2010 | Australia | Mine equipment maintenance scheduling | Mixed integer programming |
| Choi et al. | 2009 | Korea | Haulage routing optimisation of mining trucks | Multi-criteria least-cost path analysis |
| Ercelebi and Bascetin | 2009 | Türkiye | Shovel–truck dispatching | Linear programming and queuing theory |

## 3. In-Pit Crushing–Conveying (IPCC) System

The in-pit crushing and conveying (IPCC) systems are attracting more and more attention from researchers and practitioners in the mining industry, due to its advantages and benefits in comparison to the conventional ST system. The IPCC system mainly consists of the crusher and conveyor located in an open pit. The crusher is used to grind large ore blocks, and then the ground ore blocks are delivered to the surface through the belt conveyor. With the deep-mining process of an open pit, the conveyor needs to be extended while the crusher needs to be relocated at a new mining phase. An overhead view of an IPCC system in an open pit is drawn in Figure 2.

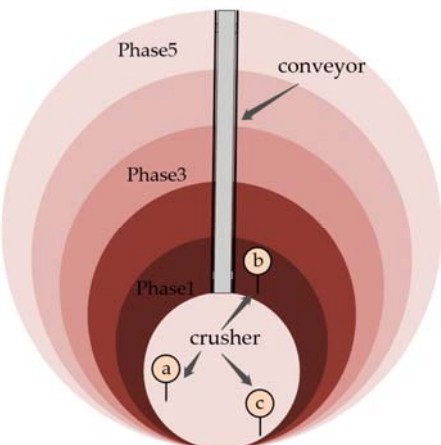

**Figure 2.** An overhead view of a sample IPCC system in an open pit in which there are one conveyor and three crushers (a–c).

In this section, an overview of recent research progress on IPCC management is presented as follows.

Gu et al. [50] introduced a new optimisation problem to determine the optimal layout of the fixed crushing station in an open-pit IPCC system. To efficiently solve the IPCC layout optimisation problem, a two-stage fusion particle swarm optimisation algorithm called TSF-PSO, which combines the merits of particle swarm optimisation (PSO) and quantum PSO (QPSO), was developed. The proposed TSF-PSO algorithm outperforms the pure PSO and QPSO in solution efficiency.

Liu and Pourrahimian [51] studied long-term production scheduling with crusher relocation in an open-pit semi-mobile IPCC system. By considering material handling and crushing station relocation costs, a MIP model was developed to determine the best location of each conveyor along with a crusher for maximizing the net present value (NPV).

Shamsi and Nehring [52] investigated the determination of the optimal transition point from a shovel–truck system to a semi-mobile IPCC (SMIPCC) system. According to the analysis of cumulative discounted costs, five different transition schemes were considered and the transition depth corresponding to the lowest economic cost was obtained.

The scheme with the lowest cumulative discounted cost was regarded as the optimal transition point.

Wachira et al. [53] evaluated the overall performance of the semi-mobile IPCC systems by the mine productivity index (MPI), which is based on equipment availability, equipment sizes and utilisation rates. The results revealed that the scenario with more than one loading equipment has a higher MPI in comparison to the scenario with only one loading equipment. In addition, haulage costs can be significantly reduced by installing the crushers inside the pits.

Paricheh and Osanloo [54] investigated the correlation between IPCC planning, open-pit mine production scheduling (OPMPS) and truck fleet sizing problems; and then developed three MIP models to integrate and solve these three problems. Compared with traditional OPMPS models without adding in-pit crushing, the integrated optimisation model reduces the fleet size and improves the NPV.

Samavati et al. [55] developed an integer non-linear programming (INLP) model to formulate the OPMPS with the addition of IPCC. To effectively solve practical-sized OPMPSP-IPCC instances, a heuristic algorithm was developed and evaluated by comparing with the exact MIP solvers (i.e., IBM ILOG-CPLEX 12.6 and Gurobi 8.0).

Hay et al. [56] indicated that the shape of a pit is heavily influenced by the ultimate pit limit. They investigated the differences and requirements of pit designing associated with the SMIPCC and ST systems. Additional constraints were considered in the new ultimate pit limit problem to return a higher NPV; and an algorithm was developed to determine the depth of crusher and the orientation of conveyor. The development of this method can determine ultimate pit limits more accurately for open-pit mines using IPCC or truck haulage.

Yakovlev et al. [57] studied the feasibility of mobile crushing units and high-angle conveyors in deep open-pit mines. Consequently, the number of mining trucks could be reduced by locating the crushing and conveying system appropriately. Such a feasibility was proved to be effective in the cyclical-and-rehandling technology by a real case study.

Abbaspour et al. [58] developed a planning tool to minimize the operation and relocation costs of a SMIPCC system. The migration of the SMIPCC system was divided into seven scenarios, each of which was analysed and selected from the perspective of both location and timing.

Paricheh et al. [59] divided the IPCC optimisation problem into two parts, namely, the design of the optimal location and the determination of optimal timing factors. First, the design problem was solved by determining the best location of the candidate crusher along with the conveyor. Then, the transportation cost of the truck fleet was compared with that of the IPCC system for solving the optimal time problem. By developing a heuristic algorithm, the performance of the proposed two-layer IPCC model was verified by a case study for a copper mine.

Paricheh et al. [60] considered the site selection and optimal relocation time of an in-pit crusher as a dynamic location problem. Two models were developed to minimise the operation and relocation costs for the IPCC system. The key parameters affecting the IPCC location were evaluated by a dynamic location–relocation IPCC case study for an open-pit copper mine.

Yarmuch et al. [61] developed discrete-time and continuous-time Markov chain models to evaluate alternative location configurations of crushers in an open-pit copper mine. By comparing the cost of purchasing new equipment with the loss caused by equipment failure rate, a discrete-event simulation model was used to verify these two models and calibrate the optimal crusher location.

Schools [62] introduced the function of online status monitoring system of belt conveyors in open-pit mines. Effective monitoring and condition monitoring systems are vital to maintaining the operation of belt conveyers and reducing potential downtime losses of the IPCC system.

Roumpos et al. [63] established a site selection model for belt conveyors in open-pit mines with the goal of minimizing transportation costs. The proposed optimisation model solved the problem of conveyor location for reducing the investment and operation cost in the life cycle of the mine. Simulation results showed that the optimisation model was robust and effective based on a case study of a lignite mine.

As in Table 1, main characteristics of recent works on the IPCC system are summarized in Table 2. According to the analysis in Table 2, some observations are made as follows. First, the number of publications on the IPCC system are much less than that of papers on the ST system, because the IPCC system is more complex than the ST system by nature. Second, most studies considered crushers and conveyors simultaneously, while studies of a single equipment type (a crusher or a conveyor) are rare. Third, as the IPCC system is a continuous system, failure (e.g., a pause) of the IPCC system will bring substantial economic losses. Moreover, the extension of belt conveyors and the relocation of crushers have a significant impact on the production safety. Therefore, most of the problem types focused on the IPCC location and performance evaluation. In comparison, the IPCC production scheduling problem is relatively sporadic. Fourth, main solution approaches for IPCC management are based on mathematical programming. The development of more efficient solution approaches such as construction heuristics and hybrid metaheuristics for optimising the IPCC scheduling problem is a promising research direction.

**Table 2.** Characteristics analysis of publications on the in-pit crushing–conveying (IPCC) system [50–63].

| Authors | Year | Country | Problem Types | Solution Techniques |
|---------|------|---------|---------------|---------------------|
| Gu et al. | 2021 | China | Layout optimisation of IPCC | Particle swarm optimisation algorithms |
| Liu and Pourrahimian | 2021 | Canada | IPCC production scheduling | Mixed integer programming |
| Shamsi and Nehring | 2021 | Australia | Optimal transition point between IPCC and ST | Analysis of cumulative discounted costs |
| Wachira et al. | 2021 | Kenya | Performance analysis of SMIPCC | Mine productivity index |
| Paricheh and Osanloo | 2020 | Iran | IPCC planning with OPMPS | Mixed integer programming |
| Samavati et al. | 2020 | Australia | IPCC production planning and scheduling | Integer non-linear programming |
| Hay et al. | 2020 | Australia | Ultimate pit limit determination for SMIPCC | Block model and network flow algorithm |
| Yakovlev et al. | 2020 | Russia | Flow diagrams of IPCC | Cyclical-and-continuous method |
| Abbaspour et al. | 2019 | Germany | Optimum location and relocation of SMIPCC | Transportation problem and scenarios analysis |
| Paricheh et al. | 2018 | Iran | IPCC location and timing problem | A heuristic approach |
| Paricheh et al. | 2017 | Iran | IPCC location problem | Mixed integer programming |
| Yarmuch et al. | 2017 | Chile | IPCC location evaluation | Markov chains |
| Schools | 2015 | USA | Condition monitoring of IPCC | Condition monitoring technology analysis |
| Roumpos et al. | 2014 | Greece | Optimal location and distribution point of IPCC | Simulation modelling |

## 4. Hybrid IPCC-ST System

Despite the rising trends in using the IPCC system, some mining companies are still hesitating to use IPCC in their mining operations due to reliability and flexibility concerns. To improve mining reliability and reduce unexpected risks, a more flexible framework is needed to make proper transition decisions between IPCC and ST systems to satisfy the location and relocation of the semi-mobile crusher.

In this section, recent works regarding hybridization or interaction of the IPCC and ST systems are discussed.

Shamsi et al. [64] presented a MIP model to determine the optimum location and relocation times of semi-mobile crushers and the production schedule of a hybrid SMIPCC-

ST system. A comparative study of ST and SMIPCC systems utilized for ore and waste handling and haulage over the life of mine showed that the SMIPCC system can greatly improve NPV by 69.77%, although the initial capital investment of SMIPCC is considerable.

Purhamadani et al. [65] estimated the energy consumption of a continuous IPCC system and a traditional truck-based haulage system and analysed the potential for energy savings from two perspectives of operating costs and energy costs. They indicated that the IPCC system is a practicable option to replace the traditional truck-based haulage system for reducing energy consumption significantly.

Bernardi et al. [66] applied a discrete event simulation model to compare the applicability of fixed IPCC, mobile IPCC, semi-mobile IPCC and shovel–truck systems. Different configurations of a pit's geometry are considered and fed into Arena (simulation software) to assess the critical parameters inherent with the operating realities of each system.

Kawalec et al. [67] studied the possibility of whether the regenerative belt conveyor system can replace the traditional truck-based haulage system in open-pit mines. By considering energy consumption and environmental protection, the actual energy demand of the regenerative belt conveyor system was modelled. They evaluated the conditioning factors to allow the replacement of a regenerative belt conveyor system, especially when the difference in transportation cost between truck-based haulage and regenerative belt conveyors is significant.

Patyk and Bodziony [68] developed a multi-criteria decision-making (MCDM) tool based on the AHP method for a surface limestone mine. By evaluating the relative weights of technological, environmental, social and economic factors, the most relevant standard for mining equipment selection was determined. The results indicated that the practical application of the MCDM tool is beneficial to support the selection of mining equipment especially in the exploitation of secondary deposits.

Krysa, Bodziony and Patyk [69] developed a model to analyse the impacts of the technological, operating and economic parameters for selected solutions and to examine the feasibility of exploiting a deposit of low quality. The model with a cyclical haulage system was applied to an open-pit limestone mine. The usefulness of the proposed model was verified in evaluating the effectiveness of each individual technological procedure and its economic aspects.

Kaźmierczak and Górniak-Zimroz [70] assessed environmental and social responsibilities for the availability of deposits based on legal, environmental and planning requirements. As a result, four deposit availability classes were introduced, namely, a very well-accessible deposit; well-accessible deposit; accessible deposit; and inaccessible deposit. A case study was carried out for 244 deposits located in Poland with the total resource amount of over 7.6 billion tons.

Chinnasamy et al. [71] developed a specific MCDM system based on the ELECTRE (ELimination Et Choix Traduisant la REalité in French and its meaning in English is "Elimination and Choice Expressing Reality") method and Dempster–Schaefer Theory (DST). The proposed DS-ELECTRE approach can reflect the advantages of DST for dealing with uncertainty while the ELECTRE method can also play a role in analysing interdependent relationships between alternatives.

Abedi et al. [72] applied the ELECTRE III method, which is a special MCDM technique, to mineral representation and integration of evidential map layers derived from geological, geophysical, and geochemical datasets. The application of ELECTRE III was validated using 3D models of Cu and Mo concentrations from 21 drill hole data.

Almeida et al. [73] conducted a multi-criteria analysis on environmental and social costs of the loader-truck and crusher-conveyor systems, respectively. Advantages and disadvantages of these two systems are compared in terms of several aspects, i.e., operating cost (without considering installation and maintenance costs), energy cost (mainly considering electricity and diesel), carbon dioxide emissions and waste control. The results showed that the crusher–conveyor system is better than the loader–truck system in the

above first three aspects, while the loader–truck system is better than the crusher–conveyor system in the last two aspects.

Ghasvareh et al. [74] applied the multi-criteria decision-making (MCDM) methods to sort out critical factors (e.g., utilisation, safety, operating cost, fuel consumption) related to the selection and design of loading and haulage equipment in open-pit mining. According to the priority of influencing factors, the MCDM methods including AHP, TOPSIS (technique for order of preference by similarity to ideal solution), AHP-TOPSIS and AHP-VIKOR were developed for the selection process.

Nunes et al. [75] applied a decision-making method to analyse the characteristics of SMIPCC and ST systems during the early stages of a mining project. Using the data available in the early mining stages as input parameters, the feasibility of each option was evaluated in terms of economic and environmental factors. In the case of a mining life cycle, the IPCC system is proved to be a more cost-effective option as it has advantages in comprehensive cost and environmental impacts.

Abbaspour et al. [76] defined the calculation formulae of safety and social indexes based on multi-variable MCDM models, for the selection of different types of mining systems, i.e., shovel–truck, fixed in-pit crushing–conveying (FIPCC), semi-fixed in-pit crushing–conveying (SFIPCC), semi-mobile in-pit crushing–conveying (SMIPCC) and fully mobile in-pit crushing (FMIPCC) systems. The evaluation results showed that FMIPCC has the highest safety index during the project life while the shovel–truck system has the highest social index.

Nehring et al. [77] emphasized the differences of mine planning approaches between IPCC and ST for hard rock operations in open-pit mines. The ST, SMIPCC and FMIPCC systems were compared in terms of various economic indicators and resource recovery rates. It was verified that the FMIPCC system can reduce the operating cost and prolong the mine life and improve the overall resource recovery.

Özfirat et al. [78] developed a fuzzy analytic hierarchy process (FAHP) method to select transportation mode in an open-pit coal mine. The proposed FAHP is based on several evaluation factors, such as transportation distance, haulage road, coal reserve, investment cost, unit production cost and production capacity. It was found that transportation distance has the highest priority among all evaluation factors, and the comprehensive performance of belt conveyor transportation is better than other transportation modes.

Rahimdel and Bagherpour [79] applied the decision-making trial evaluation laboratory model (DEMATEL) to calculate index weight; and used the TOPSIS with fuzzy set theory to evaluate three systems, namely, ST, SMIPCC and FMIPCC systems. It was concluded that the SMIPCC system could be the most suitable haulage system and the mobile in-pit crusher was better than the fixed crusher for the studied open-pit mine site.

De Werk et al. [80] conducted a comprehensive cost analysis and risk analysis of ST and IPCC systems. A Monte Carlo simulation is applied to evaluate the impact of uncertainty parameters. Their analysis showed that both mining systems are sensitive to production rate; the ST system is more sensitive to fuel prices; and the unit cost of the IPCC system is lower than that of the ST system.

Braun et al. [81] studied energy-efficient and low-emission transportation technologies that quantify sustainable development and environmental benefits. Through cluster analysis of the transportation equipment of raw materials in open-pit mine, trucks and conveyors were compared in each cluster. This study indicated that 90% of the operations are based on truck-based haulage while the remainder relies partly or completely on conveyor-based systems. In some cases, the installation of continuous conveyors instead of trucks represents a real alternative because of lower emission and operation costs. As a result, more sustainable transportation technology substitutions could be adopted especially for in-pit haulage in the hard-rock quarrying industry.

Patterson, Kozan and Hyland [82] proposed an integrated optimisation model of an open-pit coal mine with the consideration of four common open-pit coal mining subsystems: excavation and haulage, stockpiles, processing plants and belt conveyors. A MIP model

is developed to synchronize and optimise the operations of these mining subsystems for enhancing the whole-of-mine energy efficiency. Sensitivity analysis was carried out to help decision makers determine the capacities of key equipment at various mining stages.

Yakovlev et al. [83] developed a cyclical-and-continuous method to evaluate the production efficiency of the conveyor-and-truck haulage system with different capacities over specified transport distances in large open-pit mines. The efficiency of conveyor-based and truck-based haulage systems was defined when the depth of crushing stations in the system was determined. With the increase of the depth of a crush-and-reload station position, the conveyor-based haulage is proved to be more cost-effective than the truck-based haulage.

Liu et al. [84] introduced process analysis–life-cycle analysis (PA-LCA) to calculate carbon emission and energy consumption of different haulage modes in open-cut coal mines. It was verified that carbon emission will increase with the augment of slope angle. It was concluded that the performance of the conveyor-based mode is better than that of the truck-based mode in terms of energy saving, carbon emission and transportation cost.

Rahmanpour et al. [85] indicated that the hybrid IPCC-ST system is more appealing for utilisation in modern mining activities, compared to conventional shovel–truck system alone, from the perspectives of cost efficiency and operation reliability. Thus, they analysed the effective factors on determination of a suitable location of an IPCC system as a single hub by the AHP method.

Norgate and Haque [86] developed a life-cycle assessment (LCA) method to evaluate the greenhouse gas emissions during ore mining and sorting based on the equipment configuration and processing characteristics of the IPCC and ST systems. The results included that the transportation distance and annual plant feed rate affect the reduction range of carbon emission. In addition, the IPCC system has a better performance in reducing the amount of greenhouse gas.

Vujić et al. [87] discussed the characteristics of open-pit mining systems consisting of trucks, loaders, belt conveyors, spreaders, bucket chain excavators and railroads. According to the weights from high to low, six criteria were evaluated, including technology value, technology cost, technology ecological suitability, technology environmental impact, transport route applicability and level of training of the employees. The results showed that the following structure was ranked first and accepted by the mining company: bucket chain excavator–conveyor belts–spreader (ECS).

Bazzazi et al. [88] applied a fuzzy multiple-attribute decision-making (FMADM) method to deal with the open-pit mine equipment selection problem. In the FMADM method, subjective and objective attributes were defined to evaluate and compare three types of combined equipment comprising shovels, conveyers and trucks. Compared with customary decision-making methods such as FAHP, the proposed method has a better performance in equipment selection analysis.

Owusu-Mensah and Musingwini [89] applied the AHP method to evaluate four open-pit haulage modes consisting of mining trucks (contractor or mine-owned) and conveyors (surface or underground). The economic, environmental and environmental criteria of mining haulage systems were ranked according to the importance degrees. The results showed that environmental indexes account for a higher proportion in the selection process.

Table 3 concludes the main characteristics of papers on the hybrid IPCC-ST system, which contains various mining equipment types such as trucks, shovels/excavators/loaders, conveyors, and crushers. As shown in Table 3, some insightful findings are presented. First, from the perspective of problem types, evaluation factors involved on the hybrid IPCC-ST system focused on the evaluation criteria with the consideration of environmental, social, economic, reliability and safety factors. Environmental factors include greenhouse gas, harmful gas, particular substance, and waste dumps. Efficiency factors mainly concern fuel consumption of each equipment and energy efficiency of the whole mining system. Social factors contain employment rates and salary levels. Economic factors are generally related to purchasing, renting, operating and maintenance costs. Safety issues refer to

the reliability, failure rates of equipment and security of personnel. As the emphasis was placed on the performance evaluation, most papers tended to evaluate the economic value, production efficiency and environmental protection of the hybrid IPCC-ST system; but occasionally consider the system robustness, safety issues, economic factors and social indicators. Second, the majority of solution techniques for system performance evaluation are based on the multi-criteria decision-making methods. Third, due to its intrinsic complexity, the planning and scheduling optimisation methodology for the hybrid IPCC-ST system is scarce in the current literature.

**Table 3.** Characteristics analysis of publications on the hybrid IPCC-ST system [64–89].

| Authors | Year | Country | Problem Types | Solution Techniques |
| --- | --- | --- | --- | --- |
| Patyk and Bodziony | 2022 | Poland | Equipment selection in a surface mine | Multi-criteria decision-making methods |
| Chinnasamy et al. | 2022 | India | Introduction of ELECTRE for MCDM | fuzzy DS-ELECTRE |
| Shamsi et al. | 2022 | Canada | Production scheduling optimisation of hybrid IPCC-ST | Mixed integer programming |
| Krysa, Bodziony and Patyk | 2021 | Poland | Raw materials transportation | Discrete simulation |
| Kaźmierczak and Górniak-Zimr | 2021 | Poland | Accessibility of non-metallic mineral deposits | Evaluation and classification |
| Purhamadani et al. | 2021 | Iran | Energy consumption of IPCC-ST | Data analysis |
| Bernardi et al. | 2020 | Canada | Comparison of fixed and mobile IPCCs and ST | Discrete event simulation |
| Kawalec et al. | 2020 | Poland | Transition and replacement between IPCC and ST | Data analysis |
| Almeida et al. | 2019 | Brazil | ST system versus IPCC system | Environmental and economic comparison |
| Ghasvareh et al. | 2019 | Iran | Haulage system selection in open-pit mining | Multi-criteria decision-making methods |
| Nunes et al. | 2019 | Canada | Comparison analysis of SMIPCC and ST | Multi-criteria decision-making methods |
| Abbaspour et al. | 2018 | Germany | Selection analysis of ST and IPCC | Evaluation of safety and social indexes |
| Nehring et al. | 2018 | Australia | Strategic mine planning for ST and IPCC | Mine planning and evaluation |
| Özfirat et al. | 2018 | Türkiye | Selection of coal transportation mode | Fuzzy analytic hierarchy process |
| Rahimdel and Bagherpour | 2018 | Iran | Selection analysis of ST and IPCC | Multi-criteria decision-making methods |
| de Werk et al. | 2018 | Canada | Cost analysis of material handling systems | A Monte Carlo simulation |
| Braun et al. | 2017 | Germany | Sustainable technology diffusion of ST and IPCC | Data analysis |
| Patterson, Kozan and Hyland | 2016 | Australia | Integrated open-pit coal mining system | Mixed integer programming |
| Yakovlev et al. | 2016 | Russia | Conveyor-and-truck haulage system evaluation | A cyclical-and-continuous method |
| Liu et al. | 2015 | China | Energy consumption and carbon emissions of IPCC-ST | Power consumption calculation model |
| Rahmanpour et al. | 2014 | Iran | Comparison analysis of IPCC and ST | Analytic hierarchy process |
| Norgate and Haque | 2013 | Australia | Greenhouse gas impact of IPCC and ore-sorting | A life-cycle assessment method |
| Vujić et al. | 2013 | Serbia | Equipment Selection of Excavator–Conveyors–Spreader | Multi-criteria decision-making methods |
| Abedi et al. | 2012 | Iran | Analysis of mineral prospectivity mapping | ELECTRE III method |
| Bazzazi et al. | 2011 | Iran | Equipment selection of IPCC-ST | Fuzzy multiple-attribute decision making |
| Owusu-Mensah and Musingwini | 2011 | Ghana | Evaluation of ore transport options | Multi-criteria decision-making methods |

## 5. Research Opportunities

In this section, an insightful discussion on emerging research opportunities in the field of mining equipment management is presented from the perspectives of strategic-, tactical- and operational-level mining management.

At the strategic level, the entire orebody is divided into block units (e.g., millions for a large iron ore pit) in an original three-dimensional block model and the properties of each block unit (e.g., a cube of 10 m × 10 m × 10 m) are estimated by geostatistical information in the exploration stage. A preliminary task in strategic-level mining management is to determine the ultimate pit contour, i.e., the selection of block units under precedence relationship and geographical limits. In the mining literature, this type of strategic-level mining optimisation problem is called ultimate pit limit (UPIT) or mine design planning (MDP), referring to these relevant papers [13,14,16,20–24]. In the literature, the fundamental MDP was treated as a kind of "minimum-cut or maximum-flow" network flow problem in the development of efficient solution techniques [25,26].

At the tactical level, the selected block units are sequenced by assigning them into a certain number of discrete time periods. In the mining literature, this kind of tactical- level mining optimisation problem has different names as follows: constrained pit limit (CPIT) [14,90–93]; mine block sequencing (MBS) [21,94–96]; open-pit block sequencing (OPBS) [95,97–100]; open-pit mine production scheduling (OPMPS) [92,101–105]; PCPSP: precedence-constrained production scheduling problem that incorporates destination and general side constraints [14,106,107]. Indeed, these tactical-level CPIT/MBS/OPBS/OPMPS/ PCPSP problems can be transformed into a kind of "precedence-constrained knapsack or bin packing" problem if the discrete time periods are regarded as the knapsacks or bins [91,95,108–110].

At the operational level, short-term open-pit equipment planning (e.g., matching, loca- tion, assignment and dispatching of shovel–truck) and scheduling (e.g., precise timetabling of excavators or trucks with the determination of starting, processing and completion times) are drawing more and more attention, because toady's mining enterprises require more efficient control and usage of costly modern mining equipment. Recent papers on operational-level equipment planning and scheduling are still scarce in comparison to the considerable number of publications on tactical-level MBS, as referred to these papers [111–122].

Based on the above analysis, some emerging and promising research opportunities on mine equipment management at three levels are listed as follows.

- As analysed in Sections 2–4, it is rare to find academic papers on how to apply the classical machine scheduling theory (e.g., parallel-machine, flow-shop or job-shop scheduling) to model and solve the continuous-time open-pit mining equipment scheduling/timetabling problems at the operational level [21,118,123–127].
- The development of data-driven or learning-based optimisation approaches for schedul- ing is becoming a research hotspot and should be further advanced by integrating machine learning techniques (e.g., deep learning, reinforcement learning, deep rein- forcement learning, etc.) with classical optimisation methods (e.g., MIP formulation, construction heuristics and metaheuristics) to deal with the dynamic and uncertain mining equipment routing and scheduling problems in real time [22,126–139].
- Inventory (stockpiling) management with grade control is essential to mining man- agement in a volatile and demand-responsive environment. Connection between inventory control with mine equipment scheduling would be an interesting research topic at the tactical level [140–143].
- Dynamic and stochastic factors (e.g., lockdown due to pandemic, fluctuated com- modity prices, unexpected equipment breakdowns, uncertain maintenance activities, arrivals of new mining tasks) should be considered in the extended mining equipment planning and scheduling models in real-world cases [144–146].
- Selection, efficiency, productivity comparison analysis and performance evaluation of different mining systems are vital for mining practitioners. Thus, it is a promising

research direction to develop combinational qualitative and quantitative multi-criteria, multi-attribute decision-making tools for the hybrid IPCC-ST system [66,67,73,74]. Although some papers have evaluated environmental, economic and efficiency factors to select equipment of the IPCC-ST system, these factors could be considered and included in the planning and scheduling models in a multi-period multifaceted mining process [147].

- Multi-criteria decision-making (MCDM) techniques such as fuzzy AHP, DS-ELECTRA and ELECTRA III are promising to be employed for evaluating the selection of mining equipment and the feasibility of exploiting the low-quality deposits [68,70–72,148,149].
- The resource-constrained project scheduling problems (RCPSPs) with the consideration of multiple periods and various equipment types could be applied to deal with some mining equipment optimisation problems [150–153].
- Investigating how to integrate or enable interaction between the open-pit mining equipment planning and scheduling models (e.g., bi-level programming) with the whole mine-to-client supply chain procedure is worthy of more research efforts [2,5,94,154–157].
- It will be beneficial for mining enterprises to develop a serial of strategic-level, tactical-level and operational-level mining optimisation problems, models and solution approaches in an integrated or interactive decision support system [94,158].
- Mining enterprises should not only maximize profit but also fulfill their social and environmental responsibility. Resource conservation, soil erosion, fuel consumption, energy security, carbon emission, mine closure and sustainable development are prevalent topics that should be associated with mining equipment management [159–166].

## 6. Conclusions

Mining sector is an economic foundation and the main source of national wealth for many countries. Modern mining operations are ever more reliant on efficient usage of costly large-scale mining equipment (e.g., trucks, shovels/excavators/loaders, conveyors and crushers). Thus, mining equipment management is becoming crucial for the mining industry. To be viable and sustainable, mining enterprises need to operate different types of mining equipment units at various stages with the objective of minimizing the total cost or maximizing the whole productivity. In the current literature, there is a lack of a systematic and comprehensive review on mining equipment management. To fill this research gap, we reviewed over 100 recent articles relevant to mining equipment management and classified the reviewed papers into three categories: shovel–truck (ST), in-pit crushing–conveying (IPCC) and hybrid IPCC-ST systems. Based on a thorough characteristics analysis of these three categorized systems, promising research opportunities and managerial insights are discussed to inspire researchers and practitioners to develop state-of-the-art methodologies in the field of mining equipment management.

**Author Contributions:** Conceptualization, S.Q.L. and Z.L.; writing—original draft preparation: S.Q.L. and Z.L.; formal analysis, S.Q.L., Z.L., D.L. and X.L.; writing—review and editing, D.L., X.L., E.K. and M.M.; supervision, E.K. and M.M.; project administration, S.Q.L., E.K. and M.M.; funding acquisition, S.Q.L. and D.L. All authors have read and agreed to the published version of the manuscript.

**Funding:** This research was funded by the National Natural Science Foundation of China (NSFC) under Grant No. 71871064 with the title "Advanced scheduling methodologies to optimize mining operations" and Grant No. 72171054 with the title "Optimization study of industrial data-driven PCB production scheduling under uncertainty".

**Data Availability Statement:** Not applicable.

**Acknowledgments:** We would like to acknowledge the support and assistance from the CRC ORE established by the Australian Government's Cooperative Research Centers Programme, Fuzhou University, Queensland University of Technology and China University of Mining and Technology.

**Conflicts of Interest:** The authors declare no conflict of interest.

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
