# Peer review of "Recent Research Agendas in Mining Equipment Management: A Review"

_mining, doi:10.3390/mining2040043_

Round 1

Reviewer 1 Report

Only minor typos or English construction.

The contents are very valuable, as it is a thorough revision and comparison of related publications. Will be a good reference for future scholar research.

Reviewer 2 Report

Review of manuscript  mining-1915738 Recent research agendas in mining equipment management: A review

 This work proposes an extensive review on mining equipment management. As such the matter is of interest ,however the paper suffers for two serious limits:

1) There are no figures in this article, Some figures must be selected from previous literature.

2) Each part  of this manuscript only classifies and explains the original literature, but lacks further analysis of the literature, such as the second part. It is mentioned in the text that determining the best matching factor; selection with sizing of trucks and shovels ,dispatching a fleet of trucks to one shovel. you should summarize the following contents: he value of best matching factor,how to selection with sizing of trucks and shovels et al.

The third and fourth parts have the same problems

   Once the the above concerns are fully addressed, the manuscript could be accepted for publication in this journal.

Reviewer 3 Report

My Comments and Suggestions for Authors:

1. There are full of literatures listing all over the paper, including “introduction” (page 2), “ST system” (page 3-6), “IPCC system” (page 7-8), “Hybrid IPCC-ST system” (page 9-11). The paper only lists each literature and introduces what they doing in the literature. It isn’t a review, but a good reading note.

2. It lacks the analysis on the technology of mining equipment management. In each section, the studies should be classified according to the research contents, and should be discussed in detail. The paper proposes the content of each paper, but the contents of references have not relationship to each other. There should be more technology analysis such as the paragraphs after Table 1~3.

3. There could be some pictures for analysis on the key technologies in the paper, that will be friendly to readers.

4. The relationship between the research opportunities in section 5 and the former sections is not enough.

Reviewer 4 Report

The very interesting article, providing an overview of research on mining issues related to the exploitation of rock resources and mining equipment selection issues.

Please try to make a deeper research concerning actual "State of the Art". All the references seem to be relevant, but a number of them (28) are self-citations. After a short research in open access publisher's search engines, I found some interesting recent contributions (published in "Energies" and in "Recourses") that would deepen the issues presented.

Section 3 presenting mixed systems (compilation of ST and IPCC system) I would propose to add a publication presenting the problem of selection of mining technological systems consisting of ST system extended with mobile crusher and/or mobile screen often located in the mining field. Such systems are often found in European open-pit mining.

The manuscript presents research on the problem of selection of mining equipment systems depending on various parameters. Consideration should also be given to adding publications presenting research on the selection of extended ST or IPCC equipment systems, taking into account environmental, technical, technological and economic factors.

Another aspect to consider is the addition of several references showing studies on the quest for maximum utilization of the deposit with the least possible waste generation. It is proposed to review some references on the selection of the most appropriate mining equipment for the exploitation of secondary deposits and/or extraction of low-quality deposits. This technological and environmental aspect is extremely important in the design of new mines and equipment systems, as well as for existing ones.

Although the manuscript presents studies performed with various decision support tools (multi-criteria selection methods), there is no information about publications in which the ELECTRA III method was used. I believe that 1-2 articles related to the selection of mining equipment presented in this method should be added. Adding these publications will undoubtedly deepen the analysis carried out.

Editorial issues:

Line 127: de Carvalho and Dimitrakopoulos [22] - starting a sentence with a lowercase letter or a broken sentence;

Line 340: missing dot at end of sentence

Line 577: chronological presentation of literature [22,119,120-127,128].

The manuscript's presentation of issues related to the problems of mining equipment management is definitely a valuable and necessary publication that concentrates the current vision, issues and their development over the past years in one place. I believe that the manuscript should be published, having made the minor corrections presented previously.

Round 2

Reviewer 2 Report

The authors have substantially revised the paper and addressed the comments.

I suggest accept this manuscript in present form.

Reviewer 3 Report

The paper lists each literature and introduces what they doing in the literature. The studies should be classified according to the technologies, Further discussion on the technologies should be proposed in detail.

Round 3

Reviewer 3 Report

Accept in present form